# Circulating Platelet-Derived Extracellular Vesicles Are a Hallmark of Sars-Cov-2 Infection

**DOI:** 10.3390/cells10010085

**Published:** 2021-01-07

**Authors:** Giuseppe Cappellano, Davide Raineri, Roberta Rolla, Mara Giordano, Chiara Puricelli, Beatrice Vilardo, Marcello Manfredi, Vincenzo Cantaluppi, Pier Paolo Sainaghi, Luigi Castello, Nello De Vita, Lorenza Scotti, Rosanna Vaschetto, Umberto Dianzani, Annalisa Chiocchetti

**Affiliations:** 1Department of Health Sciences, Interdisciplinary Research Center of Autoimmune Diseases—IRCAD, Università del Piemonte Orientale, 28100 Novara, Italy; giuseppe.cappellano@med.uniupo.it (G.C.); davide.raineri@med.uniupo.it (D.R.); roberta.rolla@med.uniupo.it (R.R.); mara.giordano@med.uniupo.it (M.G.); beatrice.vilardo@uniupo.it (B.V.); umberto.dianzani@med.uniupo.it (U.D.); annalisa.chiocchetti@med.uniupo.it (A.C.); 2Center for Translational Research on Autoimmune and Allergic Disease—CAAD, Università del Piemonte Orientale, 28100 Novara, Italy; marcello.manfredi@uniupo.it (M.M.); vincenzo.cantaluppi@med.uniupo.it (V.C.); pierpaolo.sainaghi@med.uniupo.it (P.P.S.); 3Clinical Chemistry Unit, “Maggiore della Carità” University Hospital, 28100 Novara, Italy; 20032501@studenti.uniupo.it; 4Department of Translational Medicine, University of Piemonte Orientale, 28100 Novara, Italy; luigi.castello@med.uniupo.it (L.C.); nellodevita@hotmail.com (N.D.V.); lorenza.scotti@uniupo.it (L.S.); 5Nephrology and Kidney Transplantation Unit, “Maggiore della Carità” University Hospital, 28100 Novara, Italy; 6Immunorheumatology Unit, Division of Internal Medicine, “Maggiore della Carità” Univerisity Hospital, 28100 Novara, Italy; 7Emergency Department, “Maggiore della Carità” University Hospital, 28100 Novara, Italy

**Keywords:** circulating biomarker, extracellular vesicles, platelets, COVID-19, Sars-Cov-2

## Abstract

Sars-Cov-2 infection causes fever and cough that may rapidly lead to acute respiratory distress syndrome (ARDS). Few biomarkers have been identified but, unfortunately, these are individually poorly specific, and novel biomarkers are needed to better predict patient outcome. The aim of this study was to evaluate the diagnostic performance of circulating platelets (PLT)-derived extracellular vesicles (EVs) as biomarkers for Sars-Cov-2 infection, by setting a rapid and reliable test on unmanipulated blood samples. PLT-EVs were quantified by flow cytometry on two independent cohorts of Sars-CoV-2+ (n = 69), Sars-Cov-2− (n = 62) hospitalized patients, and healthy controls. Diagnostic performance of PLT-EVs was evaluated by receiver operating characteristic (ROC) curve. PLT-EVs count were higher in Sars-Cov-2+ compared to Sars-Cov-2− patients or HC. ROC analysis of the combined cohorts showed an AUC = 0.79 and an optimal cut-off value of 1472 EVs/μL, with 75% sensitivity and 74% specificity. These data suggest that PLT-EVs might be an interesting biomarker deserving further investigations to test their predictive power.

## 1. Introduction

SARS-CoV-2 is causing COVID-19, a pandemic burden with an unprecedented impact on healthcare systems worldwide. The clinical manifestation of COVID-19 varies from asymptomatic to a severe disease affecting the respiratory tract, that may evolve to ARDS, with patients admitted to the intensive care unit needing mechanical ventilation [1]. COVID-19 has also systemic manifestations, affecting several systems, including the cardiovascular, the gastrointestinal, the hematopoietic, the renal, and the immune one [2]. Sars-Cov-2+ patients experience several non-specific symptoms like fever, fatigue, and others. In the early phase of the infection, peripheral blood mononuclear cells are not affected, while 7–14 days after onset, when respiratory failure occurs, they are significantly decreased [2]. Moreover, at this stage, a common finding is the presence of high D-dimer levels, which are associated with a worse prognosis, possibly related to severe hypercoagulability [2]. Mild or severe cytokine storm has been described in COVID-19 patients, and interleukin (IL-) 6 seems to be the prevalent cytokine [1]. Due to the potential lethality of COVID-19 in frail patients, identification of biomarkers that may predict disease severity and prognosis is needed to guide clinical care. 

In the last years, extracellular vesicles (EVs) have attracted great interest as biomarkers for several human diseases, including viral infections [3]. EVs are lipid-bound microparticles, released by almost all eukaryotic and prokaryotic cells. Three different types of EVs are described, namely exosomes (20–150 nm), microvesicles (MVs) (100–1000 nm in diameter), and apoptotic blebs (1000–5000 nm in diameter) [4,5]. MVs are generated via a protrusion of the plasma membrane and can be distinguished for their surface antigens, deriving from the cell of origin; exosomes, are produced as components of multivesicular bodies (MVBs) and are released from cells when MVBs fuse with the cell surface, and usually express tetraspannins (CD9, CD81, CD63) as markers. EVs are transported/accumulated in biological fluids (e.g., plasma/serum, urine, breast milk, saliva, and cerebrospinal fluids or amniotic fluids) but also in solid tissues, where they deliver their contents to either neighbor or distant cells, producing physiological or pathologic effects [6].

In the blood, EVs mainly originate from resident cells and their levels increase in several infectious diseases including viral infections. This happens for Plasmodium falciparum, which is sequestered in erythrocytes and activate endothelia cells (EC) to release EVs [7]. Moreover, viruses such as the dengue virus and H1N1 influenza virus induce the release of EV from platelets (PLT) [8,9].

Subtypes of EVs, particularly MVs, can be recognized by typing specific surface markers derived from the cell that released them. In particular, PLT-EVs express platelet markers such as CD41 and CD42b [9]; leukocytes (LEU)-derived EVs express CD45 and can be released by cells of both innate and adaptive immune system; additional markers allow to identify more precisely the cell of origin (e.g., CD3 for T cell, CD11b, CD14 and CD16 for monocytes). EC may release EVs (EC-EVs) expressing the endothelial marker CD31 but also other markers, such as CD62E, CD144, CD105, and VCAM-1. EC-derived EVs play a relevant role in regulating EC survival, and are also involved in triggering the coagulation and complement cascades [10,11].

Identification and functional characterization of different types of EVs is challenging due to the lack of appropriate isolation and purification methods. Currently, differential ultracentrifugation, size-exclusion chromatography, immunoaffinity capture, and microfluidics are the options recommended by International Society for Extracellular Vesicles (ISEV) guidelines [12]. However, these might not fit diagnostic purposes, where quick and reliable methods are needed. A second point is that all these methods require several pre-analytical blood manipulation, which are stressing conditions that may promote blood cells (including PLT) to release EVs in vitro and/or may induce cell damage; these stressing conditions impact the final measurement in an unpredictable way and, this is the reason why, the translation of basic research into clinical practice, has been precluded. Among techniques for single EV analysis, flow cytometry (FC) has a high potential for clinical application due to the high throughput and multiplex fluorescence capability. However, there is a lack of consensus on standardization protocols for EV detection. Recently, it has been shown that analysis of EVs by FC can achieve a high level of reproducibility (CV < 20%) [13].

In this study, we applied FC to directly identify PLT-EVs in few microliters of fresh total unmanipulated blood [14,15,16]. This method is fast (1 h running time) and it allows to identify EVs using both a Lipophilic Cationic Dye (LCD), that binds membrane-bearing structures, and phalloidin, allowing to discriminate intact EVs from damaged ones. Intact EVs are then marked to specifically recognize and platelet-derived (PLT-CD41a and CD31) EVs and also EC-derived EVs (CD31) in the blood, since both platelets and endothelial cells express CD31 marker.

Here, we show that the pattern of circulating PLT-EVs is altered in COVID patients, which suggests that it might be used as a fast-to-test novel biomarker to guide physicians, and may be part of the diagnostic algorithm. 

## 2. Materials and Methods

### 2.1. Patients

From April 2020 to November 2020, 131 patients with suspected Sars-Cov-2 infection were enrolled at Maggiore della Carità University Hospital in Novara (Italy). Patients were hospitalized for acute respiratory symptoms or influenza-like illness symptoms during the two waves of Sar-Cov-2 infection that occurred in Italy and stratified based on the results of molecular analysis on nasal swabs (1st wave: April/May 2020, 33 Sars-Cov-2− and 23 Sars-Cov-2+; 2nd wave: October/November 2020, 29 Sars-Cov-2− and 46 Sars-Cov-2+ patients). SARS-CoV-2 infection was confirmed by reverse-transcriptase polymerase chain reaction (RT-PCR). Healthy individuals (n = 10) were enrolled as controls. The blood parameters were measured by using the Sysmex XN-2000™ Hematology System (Sysmex, Kobe, Japan) at Hospital Maggiore della Carita’ at the time of enrollment as well as EVs analysis was performed within few hours after blood withdrawal. The study was approved by local ethic committee (CE67/20); written informed consent was obtained from the patients or their legal representative.

### 2.2. Flow Cytometry Analysis

Custom EVs detection kit (Becton and Dickinson, Franklin Lakes NJ, USA) was used to characterize the EVs from the whole blood of patients. Briefly, 0.5 μL of APC-conjugated lipophilic cationic dye (LCD) and FITC-conjugated phalloidin, 5 μL of anti-CD31-PECy7 and anti-CD41a-PE were mixed in 184 μL of filtered PBS and 5 μL of whole blood was added. Phalloidin, a cyclic peptide that binds to f-actin with high affinity, was chosen to exclude from the analysis any element (apoptotic bodies or any non-intact EVs) displaying a corrupted plasma membrane [14]. This method was recently validated by Marchisio et al. [17].

Prior to staining, the reagent mix was centrifuged at 12,600× *g* for 15 min to avoid antibody’s aggregates which could be counted as EVs. After 45 min at room temperature (RT), 2 mL of PBS/2% paraformaldehyde (PFA) was added and the sample was acquired using FACSymphony A5 (Becton and Dickinson, NJ, USA). Fluorescence minus one (FMO) staining was performed for each antibody to define the gating strategy. FACSDiva software (Becton and Dickinson, NJ, USA) was used to analyze all the flow cytometry data. The count of EVs/μL was obtained using the following formula:EVs/μL = (N° of EV events for a given population*dilution factor)/(Acquired volume).(1)

The stability of flow cytometer was evaluated by acquiring four times 3 independent true count tubes (BD). We placed the threshold on fluorescent channel, which is more sensitive, as suggested in the literature [10,15]. In order to confirm specificity of LCD staining, Triton X-100 was added to the samples, which were then acquired using the same settings [16]. 

To isolate PLT, blood from three healthy donors was centrifuged at 100 g for 10 min to exclude red cells; then the supernatant was centrifuged at 2500 g for 15 min in order to pellet the PLTs, which were stained accordingly to the protocol described above.

### 2.3. Statistical Analysis

The non-parametric Kruskal-Wallis test with Dunn’s correction, Unpaired Mann-Whitney U test, and unpaired t test were used to assess the difference in demographic and clinical features between Sars-Cov-2+ and Sars-Cov-2− patients. Multivariable logistic regression model was used to evaluate the association between PLT-EVs and the probability of Sars-Cov-2 infection adjusted for sex, age, presence of comorbidities, and PLT and WBC counts. ROC curve analysis was used to determine the diagnostic value of EVs. The overall discriminatory ability of EVs was evaluated using the area under the curve (AUC). The optimal cut-off value and the corresponding sensitivity and specificity were also calculated. Leave one out cross-validation was performed to internally validate the model PLT-EVs estimating the optimism adjusted AUC and corresponding 95% CI. All tests were two-tailed and *p*-values < 0.05 were suggestive of statistically significant results. The statistical analyses were performed with GraphPad Instat software (GraphPad Software). 

## 3. Results

From April 2020 to November 2020, 131 patients with suspected Sars-Cov-2 infection were enrolled at Maggiore della Carità University Hospital in Novara, Italy. Table 1 summarizes the main laboratory findings of the patients resulting positive (n = 69) or negative (n = 62) to for Sars-Cov-2 infection. Healthy controls (n = 10) were also included. We found a significant decrease in eosinophils and basophils counts in Sars-Cov-2+ patients that also presented a reduced blood oxygenation (SpO_2_) and PaO_2_/FiO_2_ and increased blood clotting. This is in line with data showing a significant decrease of eosinophil counts at the onset of COVID-19, when compared with other type of pneumonia and a decrease of basophils linked to poor prognosis [18,19].Though both groups included old patients, we also found that the negative patients were older than the positive ones. This is in line with recent data showing that COVID19 is also affecting young adults [20]. Moreover, a different Italian study reported that the median age corresponded to 65 years [21].

The count of blood PLT-EVs was evaluated by FC. Figure 1 shows the gating strategy in detail. To confirm that the LCD specifically stains EVs, we treated the samples with Triton X-100 solution, and all signals in the LCD/phalloidin area disappeared. FMOs staining also confirmed the specificity of the signal.

We evaluated the counts of PLT-EVs in a cohort of patients enrolled during the first pandemic wave (April/May 2020) comparing hospitalized Sars-Cov-2− (n = 33) and Sars-Cov-2+ (n = 23) and healthy controls (n = 10); we found that Sars-Cov-2+ patients displayed a significant higher count of PLT-derived EVs compared to both other groups.

To confirm our findings, we also enrolled an independent cohort of patients (n = 29 Sars-Cov-2− and n = 46 Sars-Cov-2+ patients) during the second wave of COVID-19 (October/November 2020), confirming our previous results (Figure 2A). 

We also compared our method with the conventional one based on manipulation (i.e., centrifugation) of blood sample. As shown in Appendix A, we found that manipulation affects the membrane integrity of EVs, as shown by the increase of phalloidin positive events, which did not occur in our manipulation-free-protocol. 

The multivariable logistic regression model showed that PLT-EVs were strongly associated to Sars-Cov-2 even after the adjustment for potential confounding variables (*p*-value 0.0013) (Table 2).

Finally, we evaluated the diagnostic performance of PLT-EVs through a ROC curve analysis (Figure 2B) and we found an AUC = 0.79 (95% CI 0.7074–0.8648), with an optimal cut-off value of 1472 EVs/μL, with 75% sensitivity and 74% specificity. After the application of the cross-validation procedure, the optimism adjusted AUC was 0.78 (95% CI 0.6966–0.8589). These data suggest PLT-EVs to be a diagnostic marker of Sars-Cov-2 infection showing a very good diagnostic performance at the ROC curve analysis.

## 4. Discussion

COVID-19 shares some similarities with the two previous severe acute respiratory syndrome (SARS) and Middle East respiratory syndrome (MERS) outbreaks [16]. From a clinical point of view, these three infections cause fever and cough leading to lower respiratory tract disease with significant mortality especially in the elderly and in patients with previous comorbidities. Diagnosis of infection is done through molecular analysis of virus presence in throat swabs. Nevertheless, symptomatic patients who have a negative swab raise diagnostic issues and demand for easy to evaluate and reliable biomarkers. There are no predictors of clinical fate and responsiveness to therapy for SARS-CoV-2 that may support physicians, and a vaccine is not yet available. Up to date, most studies published in the literature are observational with increasing age and existing cardiovascular comorbidities identified as significant risk factors for mortality. This suggests that the identification of reliable biomarkers might be helpful to speed up clinical interventions and guide clinical care. 

In the last years, EVs received increasing attention not only for their role as mediators of cross-talk among cells, but also for their use as biomarkers for several human disorders [3]. 

In this scenario, our study provides two important results:

First, our study shows that PLT-EVs counts are higher in two independent cohorts of SARS-Cov-2 positive patients, compared to SARS-Cov-2 negative one, enrolled at the time of hospitalization, while the counts of PLTs did not vary between the two cohorts. Interestingly, a multivariate model showed that PLT-EVs are strongly associated with Sars-Cov-2 infection independently from any confonders (age, sex, comorbidities etc.). 

Our findings confirm the suggested relevance of PLT-EVs in Sars-CoV-2 infection as recently shown by Zaid et al. [22]. 

Importantly, we show that PLT-EVs is highly associated with Sars-Cov-2 infection and showed a very good diagnostic performance at the ROC curve analysis, with 75% of sensitivity and 74% of specificity. 

PLT release EVs spontaneously in response to activation or apoptosis or during long storage [23]. Noteworthy, viral infections, such as dengue virus (DV) and H1N1 influenza virus [7,9], may also play a role. DV infection activates platelets to express CD62p, CD63 and to release EVs by binding CLEC-2 [24]. CLEC-2 is expressed on the surface of PLT and ligand binding results in platelet activation and aggregation [24]. Furthermore, another study showed that DV-EVs are potent “endogenous danger signals” which trigger the activation of CLEC5A and TLR2, respectively, to promote production of proinflammatory cytokines by neutrophils and macrophages [7]. H1N1 induces surface receptor activation, lipid mediator synthesis, and release of microparticles from platelets through activation of FcγRIIA signaling [8].

Thrombocytopenia and vascular leakage are clinical hallmarks in COVID19 [25]. Interestingly, a study showed that high counts of activated PLT-EVs correlate with the occurrence of venous thromboembolis [26]. PLT-EVs bear coagulation factors and immune mediators in addition to active enzymes such as cyclooxygenase-1 and 12-lipoxygenase [27].

Second, we are here proposing a fast, cheap and easy method to detect EVs in few microliters of fresh peripheral blood by FC using a simple procedure recently patented (patent number EP3546948A1; IT201800003981A1). We followed the established flow cytometry guidelines and included in our experiments all the required controls [12,23]. We chose CD41 and CD31 as specific markers for PLT-EVs, since CD63, CD81, and CD9 are mainly exosomal markers, as recommended by the ISEV guidelines [12], and exosomes are hardly detectable by FC. Moreover, sodium citrate was selected as anticoagulant in order to minimize PLT activation in vitro and consequent EVs release, which is observed when ethylenediaminetetraacetic acid or heparin are used [28]. 

In their work, Zaid et al. [22] characterized PLT-EVs starting from manipulated blood, i.e., platelet rich plasma (PRP), after 1000 *g* centrifugation, in which PLT were subsequently removed to obtain plasma-free-platelet (PFP). PLT-EVs were then identified by FC on forward scatter (FSC) parameter, by using silica beads as a reference to assess their dimension. We observed that the global counts of PLT-EVs were numerically lower in our setting. Nevertheless, we (and others) show that manipulation of the sample (e.g., from blood to plasma) represents a stressing condition inducing blood cells, including PLT, to further release EVs and might also affect the integrity of EVs membrane, as shown by the increase of phalloidin positive events [29]. 

Brisson et al. showed that only 1% of EVs, can be detected in blood by FC when compared to electron microscopy. Nevertheless, the authors set the FC threshold on FSC parameter [30], which is far less sensitive than fluorescence. Therefore, the 80% of EVs, ranging between 50–500 nm and revealed by electron microscopy, were obviously not detected by FC, because of the setting applied. Later, the same group confirmed that FC is sensitive and reliable when the threshold is set on a fluorescent channel [31,32] as implemented in our method. 

The results of our study suggest that circulating PLT-EVs could be used as a diagnostic biomarker of Sars-Cov-2 infection, and could enter the diagnostic algorithm. Moreover, we optimized a fast and easy protocol, immediately transferable to the clinics, and identified a cut off value, which needs to be validated in larger cohorts.

Further studies are necessary to evaluate PLT-EVs variation over-time during the disease course, their power in predicting response to therapy, and to elucidate their pathogenic role in supporting venous thrombosis in a larger cohort of patients. 

## Figures and Tables

**Figure 1 cells-10-00085-f001:**
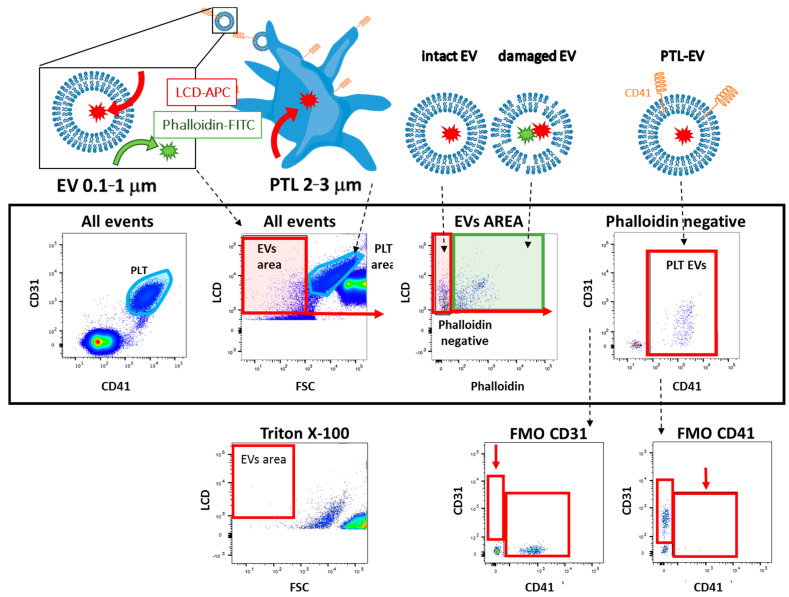
Working principle and gating strategy. The customized kit (BD) uses an APC emitting lipophilic cationic dye (LCD) (red), which diffuses into double layer structures assisted by membrane potential, thus staining both particles and cells, and Phalloidin-FITC that binds the cytoskeleton protein actin, only in particles having damaged membrane. The gating strategy is shown in the middle panel. To draw the correct EVs area, the PLT was identified from all events according their high intensity for CD31 and CD41a markers. Then, the EVs area was drawn by excluding the PLTs as shown in the second plot (LCD vs FSC). Damaged EVs were discriminated in the third plot according to the positivity to phalloidin (green) (LCD vs Phalloidin). All the LCD+/phalloidin- particles were then stained with anti-CD31 and anti–CD41a antibodies to detect platelets-derived EVs (CD41a/CD31). The lower dot plots show the negative controls; the sample was treated with a solution of Triton X-100, and then re-acquired in order to confirm the specificity of the LCD. Fluorescent minus one (FMOs) for the CD31 and CD41 are also shown. Each sample was fixed in PBS and paraformaldehyde (PFA) 2%, to avoid any further release of EVs from cells, and acquired with BD FACSymphony™ A5 and FC data were analyzed using FACSDiva software (BD, NJ, USA).

**Figure 2 cells-10-00085-f002:**
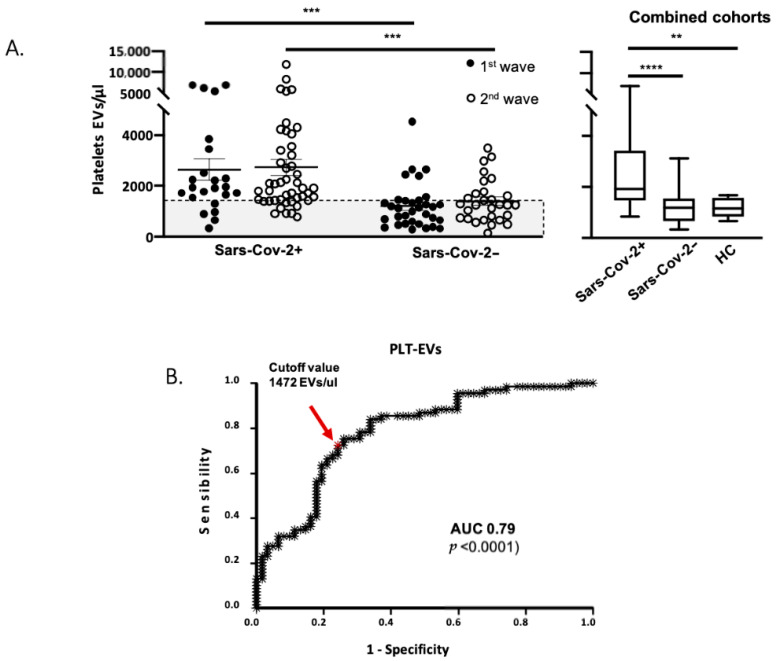
PLT-EVs count are higher in Sars-Cov-2+ patients and are an independent diagnostic predictor biomarker of Sars-Cov-2 infection. (**A**) Counts of PLT-EVs in Sars-Cov2+, Sars-Cov2− patients and HC (grey rectangle, n = 10) enrolled during the 1st wave and 2nd wave. The box plot summarizes the results from the combined cohorts. The count of EVs/μL was obtained using the following formula: EVs/μL = (N° of EV events for a given population*dilution factor)/(Acquired volume); (**B**) Combined receiver operating characteristic curve (ROC) evaluating the accuracy of PLT-EVs in Sars-Cov-2+ and Sars-Cov-2− patients. The area under the curve (AUC) was 0.79 (95% CI 0.7074–0.8648; *p* < 0.0001), the optimal cut-off was 1472 EVs/μL with 75% sensitivity and 74% specificity. Both cohorts, analyzed independently, gave overlapping results. For statistical analyses, D’Agostino and Pearson normality test was used before to perform Kruskal-Wallis with Dunn’s post-hoc test. All statistical analysis was performed using GraphPad Instat software (GraphPad Software). ** *p* < 0.01, *** *p* < 0.001, **** *p* < 0.0001.

**Table 1 cells-10-00085-t001:** Demographic and clinical features of Sars-Cov-2+ and Sars-Cov-2− patients (unpaired Mann-Whitney U test and t test were used).

	Sars-Cov-2+ (n = 69)	Sars-Cov2– (n = 62)	*p* Value
Age (years)	64 (52–75)	71 (61–82)	0.007
Gender (M/F)	45/24 (66%)	30/32 (45%)	
WBC ^1^ (×10^3^/µL)	7.8 (6.3–11.2)	9.1 (6.5–13.8)	0.07
PLTs (×10^3^/µL)	219 (149–281)	210 (155–262)	0.52
Neutrophils (×10^3^/µL)	6.1 (4–8.8)	7.1 (4.6–11.7)	0.13
Lymphocytes (×10^3^/µL)	1 (0.65–1.38)	1.17 (0.84–1.77)	0.06
Monocytes (×10^3^/µL)	0.52 (0.34–0.70)	0.58 (0.42–0.81)	0.32
Eosinophils (×10^3^/µL)	0.01 (0–0.05)	0.04 (0.01–0.16)	0.006
Basophils (×10^3^/µL)	0.01 (0–0.02)	0.03 (0.02–0.04)	>0.0001
PT-INR ^2^	1.02 (0.98–1.11)	1.07 (1.1–1.31)	0.009
APTTsec (s) ^3^	32 (28.43–35.80)	33.35 (30.03–42.35)	0.08
APTTratio	1.07 (0.95–1.19)	1.08 (0.99–1.39)	0.10
Fibrinogen (mg/dl)	565.5 (441.8–619.5)	408 (351–560)	0.06
SpO_2_	92 (86–95)	95 (93–98)	>0.0001
PaO_2_/FiO_2_	266 (217–305)	314 (260–376)	0.009

^1^ white blood cell, ^2^ prothrombin time- international normalized ratio, ^3^ activated partial thromboplastin time.

**Table 2 cells-10-00085-t002:** Multivariable model based (n = 101 subjects).

Variables	*p* Value
PLT-EVs	0.0013
Sex	0.1821
Age	0.0652
Comorbidities	0.3014
PLTs (×10^3^/µL)	0.2064
WBC (×10^3^/µL)	0.0244

## Data Availability

The data that support the findings of this study are available upon request from the corresponding author.

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
