# Peer review of "Circulating Platelet-Derived Extracellular Vesicles Are a Hallmark of Sars-Cov-2 Infection"

_cells, 2021, doi:10.3390/cells10010085_

Round 1
Reviewer 1 Report
As already stated in the first review, plt EVs are not suitable as biomarker for COVID-19 infection as there are several other pathologies associated with elevated plt EVs. The most interesting finding could be an association of increased plt derived EVs with an increased risk for thromboembolic events in COVID-19 infected patients, this should be analyzed in the patient groups.
Author Response
Rev 1
1) As already stated in the first review, plt EVs are not suitable as biomarker for COVID-19 infection as there are several other pathologies associated with elevated plt EVs. The most interesting finding could be an association of increased plt derived EVs with an increased risk for thromboembolic events in COVID-19 infected patients, this should be analyzed in the patient groups.
Answer to point 1. We thank the reviewer for his/her comment. We also agree that the association of higher count of PLT-EVs and the risk for thromboembolic events would be of clinical interest. We retrospectively looked at our data searching for major thromboembolic events i.e., pulmonary thromboembolism detected on the CT scans. It was not possible to make any evaluation in this regard since only one patient in our cohort experienced thromboembolism; a larger cohort of patients would be needed to fulfil this request.
The authors do not believe that PLT-EVs can be a biomarker per se, but rather may be part of a diagnostic algorithm including other clinic and laboratory parameters, as now stated at the end of the introduction and in the discussion paragraph.
Reviewer 2 Report
The aim of this paper is to find a new hallmark for SarsCov-2 detection.
In such regard, the high number of patients and the rapidity of the assay foster their hypothesis.
Nevertheless, there are some important aspects to take into account:
-even if one hour of time is a really short time the sensitivity at 74% and specificity at 75% with AUC 0.79 are not competitive with the classical used method so there is no real diagnostic utility.
-Authors used AUC in the combined court while to propose a diagnostic model they should have to define CD41 cut-off on the first court and then used the second one as a validation of the system
-Dimensional beads should be used in the Cytofluorimeter analysis in order to set the gate for EVs analysis.
-The authors do not perform any kind of analysis to confirm that PLTS-EVs is a specific and independent predictor of SarsCov-2.
They should have performed a multivariate statistic model in which the level of PLTS-EVs was corrected for different variables (i.e. gender, age, comorbidity….).
The increasing level of PLTS-EVs was already shown to be associated with different pathologies in different studies ( Platelet‐, monocyte‐derived and tissue factor‐carrying circulating microparticles are related to acute myocardial infarction severity. PLoS ONE. 2017;12:e0172558; Circulating endothelial and platelet derived microparticles reflect the size of myocardium at risk in patients with ST‐elevation myocardial infarction. Atherosclerosis. 2012;221:226‐231; Increased level of platelet microparticles in survivors of myocardial infarction. Scand J Clin Lab Invest. 2008;68:386‐392…..) or even associated with the use of electronic cigarettes (Electronic cigarettes containing nicotine increase endothelial and platelet derived extracellular vesicles in healthy volunteers. Atherosclerosis. 2020).
For such reasons, a multivariate model should take into account.
Author Response
Rev 2
The aim of this paper is to find a new hallmark for SarsCov-2 detection.
In such regard, the high number of patients and the rapidity of the assay foster their hypothesis.
Nevertheless, there are some important aspects to take into account:
1)-even if one hour of time is a really short time the sensitivity at 74% and specificity at 75% with AUC 0.79 are not competitive with the classical used method so there is no real diagnostic utility.
Answer to point 1: we thank the reviewer for his/her valuable comment. We agree that our test cannot compete with the current ones (molecular and antibody tests). In some people with COVID-19 disease confirmed by molecular testing (e.g. reverse transcription polymerase chain reaction: RT-PCR) weak, late or absent antibody responses have been reported. Thus, since antibody tests have limited diagnostic use for COVID19, we suggest that our method can be used as a complement to the virus detection tests for patients presenting late after symptoms onset to healthcare facilities and where virus detection tests are negative despite strong indications of infection. Moreover, the advantage of our method relies not only on the short time of procedure but also it would be less uncomfortable for the patient since it would use few microliters of blood already withdrawn during serological analyses.
2)-Authors used AUC in the combined court while to propose a diagnostic model they should have to define CD41 cut-off on the first court and then used the second one as a validation of the system.
Answer to point 2: We agree with the reviewer on the need to validate the diagnostic model; however, the use of the first cohort as training and the second as validation would reduce the sample size leading to highly variable results. Therefore, we applied a cross-validation procedure with the aim to validate the model without reducing the overall sample size allowing the calculation of the optimal adjusted AUC and corresponding confidence interval that are reported and in the current version of the manuscript.
3)-Dimensional beads should be used in the Cytofluorimeter analysis in order to set the gate for EVs analysis.
Answer to point 3. In our setting, this control is amendable since Marchisio et al. (doi.org/10.3390/ijms22010048) validated the EV-kit using dimensional beads to identify the EV compartment on the basis of size values, therefore distinguishing EVs from other blood elements (i.e. PLT and red cells). We replicated what suggested by the authors: we set the gates using PLTs as dimensional reference. and we also included negative controls, i.e triton X-100 to define the EVs area and single FMOs in order to design the specific gates for CD31 and CD41 markers. As shown in Figure 1 after the membrane disruption (Triton X-100 treatment), the whole EV population (LCD+/Phalloidin- events) disappeared.
4) -The authors do not perform any kind of analysis to confirm that PLTS-EVs is a specific and independent predictor of SarsCov-2. They should have performed a multivariate statistic model in which the level of PLTS-EVs was corrected for different variables (i.e. gender, age, comorbidity….).
The increasing level of PLTS-EVs was already shown to be associated with different pathologies in different studies ( Platelet‐, monocyte‐derived and tissue factor‐carrying circulating microparticles are related to acute myocardial infarction severity. PLoS ONE. 2017;12:e0172558; Circulating endothelial and platelet derived microparticles reflect the size of myocardium at risk in patients with ST‐elevation myocardial infarction. Atherosclerosis. 2012;221:226‐231; Increased level of platelet microparticles in survivors of myocardial infarction. Scand J Clin Lab Invest. 2008;68:386‐392…..) or even associated with the use of electronic cigarettes (Electronic cigarettes containing nicotine increase endothelial and platelet derived extracellular vesicles in healthy volunteers. Atherosclerosis. 2020).
For such reasons, a multivariate model should take into account.
Answer to point 5. We thank the reviewer for his/her comment. To address this point, we recruited a statistician (Dr Lorenza Scotti) who helped us in performing a multivariate model. The multivariable logistic regression model showed that PLT-EVs are strongly associated to SarS-Cov-2 infection, even after the adjustment for potential confounding variables (comorbidities, age, sex etc). These results are shown in the new Tables 2.
Reviewer 3 Report
The authors have followed my suggestions, added data and improved the text.
Author Response
Thanks for the kindly review.
Round 2
Reviewer 1 Report
No further comments.
Reviewer 2 Report
thanks for the answers
This manuscript is a resubmission of an earlier submission. The following is a list of the peer review reports and author responses from that submission.
Round 1
Reviewer 1 Report
This manuscript is very timely and publishable. It requires just a couple of minor improvements.
The cytometry method applied is not the standard approach. The authors should discuss potential technical issues - what is the technical variability? What was the day to day variability? What were the numbers of total EVs identified? What were the sizes of these EVs? where these exosomes?
Interestingly, the SARS2 group was 10 years younger than the SARS2 negative group. The reasons are not clear to me since high age is the main risk factor for severe Covid. Regardless the reasons... the interpretation is affected. The most severe Covid is most common among patients older than 80 years who were not included in the study. The authors can either collect more samples from elderly SARS2 positive patients or do a multifactorial analysis with age as confounding factor.
In addition to age which surely needs to be included in the statistical analysis, WBCs and neutrophils are nearly significantly different. This can be included in the Discussion as potential source of bias.
please, do not use more than 2 decimal spaces.
The EVs were not analyzed dynamically, so, the authors should not use the term "increase", but rather explain that a group had higher EVs. This is of importance, since auch associations do not have to be causative and even if they are, the direction is not clear. It might be that the altered counts of EVs are causing higher risk for SARS2 infections or severe Covid.
What was the correlation of EVs and hypoxia or other functional parameters of the lung?
Although it is rare that the patients have SARS2 positive blood, where these patients analyzed in this respect?
In which phase of their Covid were these patients?
Reviewer 2 Report
Review comments
Title: Circulating platelet-derived extracellular vesicles are 2 a potential hallmark of Sars-Cov-2 infection
Authors: Giuseppe Cappellano, Davide Raineri, Roberta Rolla, Mara Giordano, Chiara Puricelli, Beatrice Vilardo, Marcello Manfredi, Vincenzo Cantaluppi, Pier Paolo Sainaghi, Luigi 5 Castello, Nello De Vita, Rosanna Vaschetto, Umberto Dianzani and Annalisa Chiocchetti.
In this study, Cappellano et al explored a new approach to analyze circulation EVs from a very small amount of blood without laborious isolation protocols and then analyzed blood from patients with SARS-Cov-2-infection, patients with other cause of respiratory distress, and healthy controls. The motive for the study is good, and the need for faster methods for EV analytics in clinical use is true. Although the study shows some promise and can show an increase of platelet origin material in SARS-Cov-2-patients, it falls short to convince me that the increase is due to the EVs. Authors need to prove with other EV methods that the reported difference is coming from increased concentrations of EVs.
Line 59-61: I would suggest a better reference for this sentence, like Raposo G, Stoorvogel W. Extracellular vesicles: exosomes, microvesicles, and friends. J Cell Biol. 2013 Feb 18;200(4):373–25. PubMed PMID: 23
Line: 91-92 the authors state that phalloidin was used to discriminate intact EVs from damaged ones. Phalloidin is not usually used in EV flow cytometry and I would like to know the logic of how this was chosen. Were the phalloidin-positive material of the same size as phalloidin-negative (presumable EVS)?
Line 105-116: What was the size threshold of the used FACS? Were any beads used for size determination? Are the authors sure that the measured events are in fact EVs? Many conventional flow cytometers have a threshold of around 300 nm, which of course then allows the detection of only big EVs.
Several aspects of the flow cytometry analysis need to be clarified and presented:
- Did the authors check does the fluorescent labels, even after filtration, give any signal?
- Did the authors do mock labeling of blood, and did it give background?
- Did the authors use any positive control for the study, e.g. isolated EVs to show that the system works?
- To ensure that the detected signal is from EVs, authors need to verify it with adding a detergent to the samples and re-run them (see Osteikoetxea, Xabier, et al. "Differential detergent sensitivity of extracellular vesicle subpopulations." Organic & biomolecular chemistry 13.38 (2015): 9775-9782.
- Gating strategy in detail?
- How many events were collected?
Table 1: Was the blood parameters measured with the same instrument for all the participants? Were there delays in sample preparation and analysis?
Line 174-175: the reference used here points out only to the viral infections, not “several human disorders”. revise.
Line 177-181. To authors to make so bold claim that they show an increase of platelet-derived EVs in patients with SARS-Cov-positive patients, the evidence is thin. This assumption is based on a FACS experiment that is not very convincing. I think that the authors have a good idea, and they show that there is an increase of platelet origin material, but without further experiments, there is no way to say whether platelet-derived EVs are analyzed here. Did the authors try to use an EV marker in the study?
Line 195-202: The method presented here using only a few microliters of blood does have value in clinical settings, but more experiments need to be done to validate that the method is suitable for EV detection. As already shown in a paper by Arraund et al (Extracellular vesicles from blood plasma: determination of their morphology, size, phenotype, and concentration), only around 1 % of EVs in circulation can be detected with FACS. Thus, the comparison of the EV concentration in this manuscript and Arraund et al raises more questions. The choice of anticoagulant is correct and well thought of.
Reviewer 3 Report
This paper proposes that PLT-derived EVs might be an interesting biomarker to identify Sars-CoV2 positive patients.
The strength of the presented method is that the authors show an easy and fast procedure to identify the number of EVs (1h running time).
Despite this, the one and only experiment with flow cytometry is not sufficient to conclude that the Sars-CoV2 positive patient has more EVs derived from PLT.
-Patients included in the study are all hospitalized. The timing of the blood sample collection needs to be included and should be at the time of hospitalization. This information is mandatory because different therapy/treatment could drastically affect the readout of the study.
-The authors used an APC-lipophilic dye to stain EVs. Is important to consider that in the whole blood different type of EVs are present including lipoprotein (that has the same size of Extracellular Vesicles). Furthermore, is plausible that during Sars-Cov2 infection cells that undergo apoptosis release an increasing number of apoptotic bodies that could be found in the blood. For such reason, the type of EVs analyzed should be minimally characterized (western blot or flow cytometry for EVs markers, NTA...). Is perfectly understandable that a diagnostic test could not include a pre-purification analysis but to confirm the specificity of the methods the authors should show the same assay performed on purified EVs.
-Size control beads should be used to gate the flow cytometry in order to identify the "EVs area"
-In order to reinforce the message that the CD41+ vesicles are extracellular vesicles double staining with an EVs marker (like CD63 or CD81 or CD9) should be performed. That does not change the aim of a fast and easy diagnostic method but is essential to say that the authors are measuring EVs.
-In order to exclude selection bias and to asset the generalizability of the proposed model the same cut off should be tested in an external validation court.
-More clinical information about the patients should be included.
Reviewer 4 Report
The authors focus on circulating extracellular vesicles (EVs) in patients with or without COVID-19 infection and state that patients with COVID-19 infection show increased numbers of platelet-derived EVs. They conclude that this observation may enable discrimination between these groups of patients using platelet-derived EVs as biomarkers. Although the topic is highly interesting, this study has several limitations as there are ongoing efforts in the Scientific community to define international standards for isolation, purification and characterization of EVs and to enable comparability of experimental outcomes, e.g published in position papers of the ISEV (MISEV2018, doi: 10.1080/20013078.2018.1535750).
A major critical issue is that the authors completely ignore these recommendations. The simple analysis of EVs from whole blood by flow cytometry is not appropriate and has the disadvantages of contaminating blood cells, debris and lipoproteins. This is also in contrast to ref 14 and 15, where the authors analyzed EVs from cerebrospinal fluid.
There is no information about pre-analytic conditions to collect and store the blood samples from the patients and controls. There is no further purification of the analyzed EVs and no parameters analyzed for EV characterization.
Furthermore, these EVs are not suitable as biomarkers for COVID-19, there are PCR tests available to test for the virus. Many other causes are reported for an increase of plt-derived EVs and these risk factors for a procoagulant state have to be matched to compare patients. An increase of plt derived EVs may be a biomarker for an increased risk for thromboembolic (TE) events in patients with COVID-19 infection. However, this would require a prospective study of plt-derived EVs in COVID-19 patients with and without TE.